# Fusion-pMT: Biological Language Modeling for Tri-Molecular Binding in Immunogenicity Prediction

## Abstract

Recent advancements in multimodal techniques and large language models (LLMs) offer a new perspective on handling biological sequences through biological language modeling. One particularly critical yet underexplored challenge lies in modeling the tripartite interaction among peptide, MHC, and TCR—an essential step in understanding T cell-mediated immunity and improving immunogenicity prediction. In this paper, we propose **Fusion-pMT**, a biological language modeling framework that (1) learns unified representations of the three molecular inputs by leveraging their common structure as amino acid sequences, and (2) fuses the representations of each sequence to enable interaction among heterogeneous molecular inputs, aligning with the stepwise nature of immune recognition. Built on this foundation, Fusion-pMT effectively supports both pairwise and tripartite interaction modeling among peptide, MHC, and TCR. Moreover, its parameter-sharing design reduces memory usage during inference, making it lightweight and practical for biological applications. To validate its effectiveness, we conduct comprehensive experiments covering both pairwise and tripartite interactions (including out-of-distribution evaluation) and demonstrate that Fusion-pMT consistently outperforms state-of-the-art baselines across all the benchmarks.

## 1 Introduction

The success of multimodal techniques and large language models (LLMs) has demonstrated the remarkable ability of Transformers to process diverse sequential data, including biological languages (Ji et al., 2021). This breakthrough has catalyzed a growing body of research in the data mining and machine learning community to model molecules through bio-sequence representations (Pei et al., 2024; Park et al., 2024), paving the way for deeper explorations of complex biomolecular interactions in immunology.

Among these problems, one of the critical challenges is understanding how biological sequences dictate immune recognition—particularly the intricate interactions among the ❶ antigenic **peptide**, ❷ major histocompatibility complex (**MHC**), and ❸ T cell receptor (**TCR**). Although structural data can provide valuable insights into molecular recognition, annotated peptide–MHC–TCR complex structures are extremely scarce, and AlphaFold (Jumper et al., 2021) struggles to accurately model these complexes without docking priors (Bryant et al., 2022; Bradley, 2023; Chen et al., 2024). These limitations motivate us to continue exploring immunogenicity prediction from a language perspective. In particular, immunogenicity prediction can be viewed as an immunological conversation, where the core principle is self–foreign discrimination (Wortel et al., 2020). The distinction between non-self and self peptides parallels the distinction between foreign and native languages (Vu et al., 2024), as illustrated in Figure 1a.

Although a number of prior studies have investigated immunogenicity interaction prediction (Montemurro et al., 2021; Chu et al., 2022; Yang et al., 2023), their discussions were limited to two of the three molecules, i.e., pairwise interactions. This limited scope restricts their clinical applicability, both from a biological perspective—which requires capturing the complete biological process—and from a computational standpoint, where the law of total variance (Weiss et al., 2006) emphasizes the need to account for all sources of variation. This further highlights the urgent need to model the full tripartite interaction, especially given the increasing

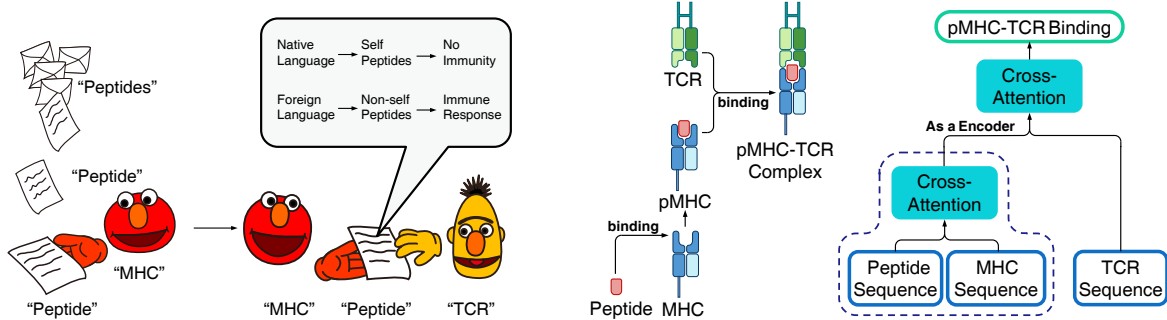

(a) Immunological Conversation.    (b) Biological Process vs Model Architecture.

Figure 1: The role of pMHC–TCR in adaptive immunity and the correspondence between our model architecture and the biological process. (More details will be introduced in Section 3.1.) **(a)** The immunological conversation. The self–foreign discrimination relies on the sequences of immunological molecules. **(b)** The model training workflow dissects peptide-MHC binding and pre-trains the corresponding module for subsequent pMHC–TCR binding predictions (right panel), mimicking the real biological process (left panel). The dashed box indicates the module that requires pre-training.

demand in areas like designing cancer vaccines (Rojas et al., 2023; Yarchoan et al., 2024) and guiding personalized immunotherapy (Hassel et al., 2023; D'Angelo et al., 2024; Mullard, 2022).

In response to this critical problem, pMTnet (Lu et al., 2021) is one of the earliest works to explore the interaction between peptides, MHC, and TCRs for immunogenicity prediction. However, despite the significant length differences in TCR sequences, their approach represents the TCR as a separate vector, distinct from the other two sequences, leading to suboptimal modeling of the complex trimolecular binding. Built on this work, PISTE (Feng et al., 2024) directly encodes the three sequences using a sliding-attention transformer. However, neglecting the underlying biological process raises concerns that this design may compromise performance.

Guided by the goal of improving biological alignment for trimolecular binding (Molnar, 2020), we design a model that ① preserves sequence-specific characteristics and ② mirrors the biological binding process. Specifically, we introduce a strong architectural inductive bias that enforces the hierarchical dependency of immune recognition—peptide-MHC binding must precede TCR interaction—thereby solving the information bottleneck inherent in traditional late-fusion models. For ①, inspired by LLMs that process variable-length sequences while retaining structure (Devlin et al., 2019), we adopt the sequence modeling techniques within for capturing peptides, MHCs, and TCRs. We further employ a shared embedding layer and encoder for Fusion-pMT to learn unified amino-acid representations across all three inputs, capturing cross-molecular dependencies and improving immunogenicity prediction. Removing per-molecule encoders reduces parameters by about two-thirds, cutting storage and memory usage and simplifying the architecture. For ②, we follow the two-step activation in Figure 1b: (a) peptide–MHC binding to form pMHC (Kammertoens & Blankenstein, 2013), then (b) TCR binding to pMHC (Huppa et al., 2010). A new multimodal fusion module we propose respects this order, aligning the model with the biological process.

Overall, leveraging representation learning approaches from language modeling, we present **Fusion-pMT**, which integrates information from all three sequences and offers a template for modeling richer biological "languages." We validate its efficacy through extensive experiments and ablations on real-world datasets, covering pairwise and tripartite interactions. In summary:

- We propose **Fusion-pMT** for peptide–MHC–TCR triad binding that preserves sequence form and aligns the model with biological steps.

- We introduce unified token embeddings and a step-aware multimodal fusion for sequence integration, improving immunogenicity prediction.

Table 1: Comparison of common models for immunological sequence binding, highlighting differences in model components and concatenation methods.

| Model | MHC Modeling | Peptide Modeling | TCR Modeling | Binding Mechanism |
|---|---|---|---|---|
| STMHCpan (2023) | Peptide–MHC Graph | Peptide–MHC Graph | N/A | Star-Transformer |
| TransPHLA (2021; 2022) | Self-Attention | Self-Attention | N/A | Self-Attention |
| PISTE (2024) | Self-Attention | Self-Attention | Self-Attention | Sliding-Attention |
| netMHCpan (2024) | LSTM | LSTM | N/A | Concat |
| CcBHLA (2023) | BiLSTM | BiLSTM | N/A | CNN |
| ESM-2 (2023) | Transformer | Transformer | Transformer | Concat |
| UniTCR (2024) | N/A | N/A | Self-Attention | Cross-Attention |
| DeepAIR (2023) | N/A | N/A | Self-Attention | Gate-Based Attention |
| NetTCR (2021) | N/A | 1D CNN | 1D CNN | Concat |
| pMTnet (2021) | LSTM | LSTM | Autoencoder | Concat |
| **Fusion-pMT (This work)** | **Shared Embedding** | **Shared Embedding** | **Shared Embedding** | **Cross-Attention** |

- We empirically ablate and evaluate these components, demonstrating effectiveness, versatility, and practical relevance.

## 2 Related Works

The prediction of interactions among peptides, MHC, and TCR is crucial in immunoinformatics. However, most methods focus on TCR–antigen specificity or peptide–MHC class I binding, with only a few addressing peptide–MHC–TCR triad binding due to its complexity and the scarcity of experimental data. We identify immunological language modeling and molecular binding mechanisms as key factors and review related work accordingly (summarized in Table 1 for the reader's convenience). In addition, we discuss the opportunities and challenges of incorporating structural information into pMHC–TCR modeling.

**Immunological language modeling.** Previously, traditional methods represented immunological sequences in non-sequential forms. For instance, Montemurro et al. (2021, NetTCR) employed CNN encoders for TCR and antigenic peptides, and models such as DeepAttentionPan (Jin et al., 2021) and CapsNet-MHC (Kalemati et al., 2023) enhanced traditional CNNs with attention layers to improve feature extraction. With advances in representation learning, sequence modeling techniques have gained popularity. TransPHLA (Chu et al., 2022; 2021) incorporated self-attention modules to capture complex dependencies. STMHCpan (Ye et al., 2023) instead modeled peptide–MHC interactions as graphs, introducing graph neural networks to the field.

**Molecular binding mechanisms.** NetTCR (Montemurro et al., 2021) employed direct concatenation of hidden embeddings, while TransPHLA (Chu et al., 2022) utilized self-attention. UniTCR (Gao et al., 2024) integrated RNA sequence data with TCR analytics via cross-attention, though its clinical relevance is limited by not accounting for the peptide–MHC binding prior to TCR interaction.

**Models for Peptide–MHC–TCR Triad Binding.** pMTnet (Lu et al., 2021) is the first model proposed for directly modeling peptide–MHC–TCR triad binding. This model relies heavily on two pre-trained modules, netMHCpan (Jurtz et al., 2017; Reynisson et al., 2020) and Tessa (Zhang et al., 2021), to encode the three sequences. Specifically, it employs a vector concatenation strategy to model TCR–pMHC interactions. PISTE (Feng et al., 2024) further introduced the direct feeding of the three sequences into a sliding-attention transformer for prediction, while leveraging sliding-attention to capture physics-driven dynamics. Nevertheless, this approach neglects the two sequential steps of the real biological process involved in the immune response, compromising its biological alignment.

**Incorporating Structural Information: Opportunities and Challenges.** While combining sequence and structural information is generally advantageous in protein modeling, structural annotations for pMHC–TCR complexes are extremely scarce (≈350 known 3D structures (Kaas et al., 2004); covering 1% of our sequence data). Furthermore, to the best of our knowledge, only a single study (Bradley, 2023) has

attempted to incorporate structural information into tripartite immunogenicity prediction. They used AlphaFold 2 (Jumper et al., 2021) to predict pMHC–TCR complex structures, with 12 known docking templates guiding the folding process and thereby introducing strong structural priors. Given that the method was validated on 130 samples, its applicability to broader scenarios remains uncertain. Although structure prediction tools such as AlphaFold (Jumper et al., 2021; Abramson et al., 2024) have achieved remarkable success in modeling individual protein structures, accurately capturing trimolecular pMHC–TCR complexes remains a challenge. In this study, we therefore focus on *sequence-level modeling* across peptide, MHC, and TCR sequences, aiming to prioritize biologically meaningful candidates for downstream immunological investigation and targeted experimental validation.

## 3 Preliminaries

In this section, we first outline the core molecules of the immune system in Section 3.1, then discuss the computational challenges of multi-sequence biological problems in Section 3.2, and finally introduce the cross-attention mechanism in Section 3.3.

### 3.1 Biological Sequences in Immune Systems

Adaptive immunity hinges on two key processes: **antigen presentation** and **antigen recognition**. Antigen-presenting cells (APCs) first bind antigenic peptides to MHC, forming peptide-MHC (pMHC) complexes, and then T cells **recognize** these complexes via the T-cell receptor (TCR), forming pMHC–TCR complexes (Huppa et al., 2010). T cells harbor highly diverse TCRs composed of $\alpha$ and $\beta$ chains, wherein $\beta$-chain diversity is central to distinguishing self from non-self antigens (Mora & Walczak, 2019).

Consequently, successful immune responses require both MHC-mediated presentation and TCR-mediated recognition. A broader overview of immune molecules is provided in Appendix B.1, and the embedding methods of amino acids are deferred to Appendix B.2.

### 3.2 Molecular Binding Tasks

Research on immunological sequence binding highlights several challenges: ❶ **Peptide-MHC binding** is crucial for antigen presentation, and useful for vaccine development. However, not all peptides binding to MHC can form a pMHC complex that also binds to TCR, limiting the model's reflection of entire cellular immunity. ❷ **Peptide-TCR binding** is critical for T-cell activation and T-cell therapies. However, we notice Peptide-TCR binding requires a suitable MHC, which is missed in this task. Therefore, we specifically dismiss this task. ❸ **Peptide-MHC-TCR binding** is vital for understanding cellular immunity, and useful for vaccine development. It offers a holistic view of immune recognition, facilitating better disease-combating strategies. Our paper is therefore committed to this holistic challenge ❸, and a quick adaptation can be applicable to task ❶ (see the discussion of Fusion-pM / Stage 1 in Section 4.3) due to the alignment of our method with the real biological process.

In general, for the input sequences $\boldsymbol{X}_p, \boldsymbol{X}_M, \boldsymbol{X}_T$ (corresponding to peptide, MHC, and TCR sequences respectively), the binding task ❸ can be formulated as modeling the following probability:

$$\mathbb{P}\left\{\boldsymbol{X}_p, \boldsymbol{X}_M, \boldsymbol{X}_T \text{ altogether trigger an immune response}\right\},$$

which matches the modeling in language tasks and justifies the usage of representation learning techniques within. Similarly, task ❶ induces the modeling of the probability

$$\mathbb{P}\left\{\boldsymbol{X}_p, \boldsymbol{X}_M \text{ form a pMHC complex}\right\},$$

which corresponds to "antigen presentation" and is the necessary step for the tri-molecular binding process. Moreover, real-world immunogenicity experiments provide non-binary and probabilistic assessments as supervised signals, different from regular deep learning tasks; nonetheless, common cross-entropy loss can already cater such irregular labels, considering that cross-entropy loss refers to the KL divergence between the label distribution (i.e., non-binary and probabilistic assessments) and the model estimation distribution.

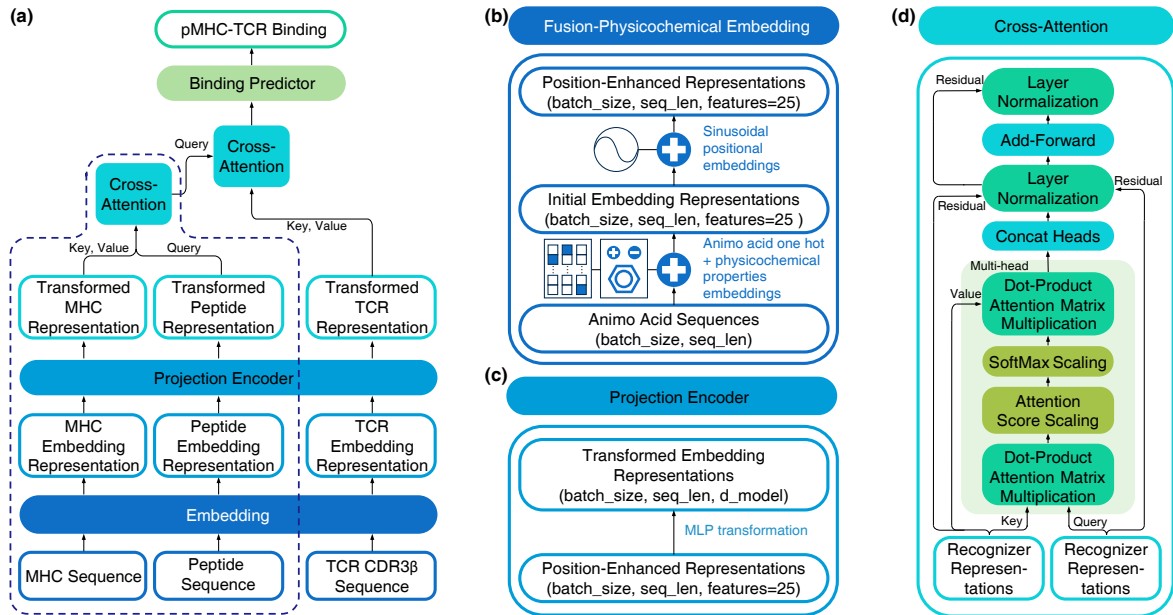

Figure 2: An overview of our model structure. The figure illustrates our complete model architecture for remodeling peptide–MHC–TCR triad binding as a representation learning and sequence fusion task. Notably in **(a)**, the modules in the dashed box that involve peptide and MHC are first pre-trained on a peptide–MHC binding task (cf. the details in Section 4.3), which provides a robust initialization prior to the core fine-tuning (which concerns all the modules in **(a)**) for pMHC–TCR binding predictions. Additionally, **(b)** illustrates the "Fusion-Physicochemical Embedding" module, **(c)** demonstrates the "Projection Encoder" module, and **(d)** depicts the "Cross-Attention" module. These components collaboratively enhance feature extraction and cross-sequence interactions.

For the reader's convenience, Table 4 in Appendix B.1 provides a comprehensive overview of the key molecules involved in antigen presentation, detailing their cellular locations, structural properties, primary functions, theoretical diversity, and sequence homology.

### 3.3 Cross-Attention

Cross-attention (Hou et al., 2019; Chen et al., 2021) gains prominence in various sequence interaction tasks, including text translation (Gheini et al., 2021), image captioning (Zhang et al., 2023), and voice recognition (Sun et al., 2021). Its core advantage lies in enabling one sequence to selectively attend to relevant parts of another, thus enhancing interaction modeling (Ju et al., 2021; Jin et al., 2023). This characteristic happens to parallel the biological selectivity and specificity of immune responses, aiding in accurate prediction of binding affinities and antigen presentation (Kurata & Tsukiyama, 2022).

Formally, a cross-attention module is given as:

$$\mathrm{Attn}(\mathbf{Q}, \mathbf{K}, \mathbf{V}) = \mathrm{softmax}\Big(\mathbf{Q}\mathbf{K}^\mathsf{T}/\sqrt{d}\Big)\mathbf{V}, \tag{1}$$

where $\mathbf{Q}$, $\mathbf{K}$, and $\mathbf{V}$ are the query, key, and value matrices derived from the input sequences with dimension $d$. Following Chen et al. (2022a), $\mathbf{K}$ and $\mathbf{V}$ typically **originate from the same sequence**, enabling the model to align context across biological sequences such as TCRs, MHCs, and antigens.

## 4 Remodeling Peptide–MHC–TCR Triad Binding as Sequence Fusion

To comprehensively understand and predict peptide–MHC–TCR interactions, accurate representation of protein sequences is indispensable. This section delineates our approach to capturing both the spatial relations

among distinct sequences (Section 4.1) and the inherent characteristics of amino acids (Section 4.2). Through the model proposed in Section 4.3, we aim to preserve the innate sequential characteristics of proteins, which are crucial for understanding their biological functions and interactions. Important implementation details are discussed in Section 4.4.

## 4.1 Representing Biological Sequences

In this subsection, we outline protein sequence representation methods used within our model, emphasizing the critical need to accurately capture both the amino acids and their positional information. In general, our approach preserves the intrinsic sequential nature of biological sequences throughout the encoding and transformation processes; the comparison with the traditional vector representation is presented below, and more details on positional encoding and context-aware embeddings are provided in Appendix A.

**Issues with a vector representation.** The transformation of protein sequences into vector representations, as adopted in Lu et al. (2021) and other pioneering works, poses several challenges. One major issue is the **potential loss of sequential context and structural information**, which are critical for understanding protein functionality. Traditional vectorization methods often flatten the sequence, treating it as a mere collection of features without considering the natural order and interactions between amino acids. This results in significant information loss, particularly in cases where the spatial arrangement and chemical properties of amino acid residues dictate their interactions and functions. A notable limitation of pMTnet (Lu et al., 2021) is its reliance on LSTM-based encoder, which, while effective for capturing local sequential dependencies, struggles to preserve long-range dependencies and structural interactions crucial for binding prediction.

**Importance of Sequence Form.** The structural form of a protein sequence–its sequence of amino acids and their respective positions–plays a pivotal role in determining its biological function. Proper representation of these sequences is crucial for computational models to predict protein interactions. Our method emphasizes the maintenance of the sequential integrity of protein sequences to ensure that both local and global structural characteristics are accurately represented, which is essential for predicting interactions. As **an empirical justification**, we verify the benefit of maintaining the sequence form for biological sequences through ablation studies in Section 5.5.

## 4.2 Unified Encoders for Heterogeneous Sequences

In a representation learning study, Chen et al. (2022b) argued that similar representations can enable more effective use of the attention mechanism. Following this observation, we accordingly suggest that MHC and antigenic peptide sequences share the same encoder, so that the cross-attention mechanism we propose can be more effective in modeling sequence fusion.

In more detail, for the one-hot embedding matrix of a sequence $X$, whether it corresponds to a peptide, MHC, or TCR, **the linear transformation matrix $W$ is identical** (as shown in Figure 2(a)). We note that this technique enforces the same encoding for different biological sequences, which aligns with the real binding process, considering that the underlying amino acids are identical across biological sequences.

## 4.3 Complete Learning Mechanisms

We incorporate the representation learning techniques above and present the full process of our proposed model, Fusion-pMT, which is further elaborated in Figure 2. On the learning side, we follow a common "pre-training + fine-tuning" paradigm to handle the three input sequences; on the architecture side, we first fuse the peptide and MHC and then fuse with the TCR sequence, which not only resembles the biological process but also effectively utilizes the abundant data for peptide–MHC interactions. Specifically, in Stage 1, we pre-train the peptide and MHC modules, yielding a model we denote as Fusion-pM, which can already serve as a standalone predictor for the peptide–MHC binding task; in Stage 2, this backbone is extended by incorporating the TCR sequence, resulting in the full Fusion-pMT model for trimolecular binding prediction.

**Stage 1: Pre-training via peptide–MHC binding.** As depicted in Figure 2(a), we first train only the modules in the dashed box that involve the peptide and MHC sequences on a peptide–MHC binding prediction task. Specifically, we use pre-training data (peptide and MHC sequences along with their binding labels) from Chu et al. (2022) and feed these sequences into the partial model.

As illustrated in Figure 2(b), we first encode the amino acid sequences with one-hot and physicochemical property embeddings (Yang et al., 2018). The model then transforms peptide and MHC sequences into high-dimensional embeddings via linear transformations, and further incorporates structural information via sinusoidal positional encodings (cf. Appendix A.2), which help preserve sequence integrity and temporal dynamics.

The projection encoder (illustrated in Figure 2(c)) further lifts the sequence dimension to $d$ (so that it aligns the dimensionality of different sequences) through an MLP module and provides extra flexibility. Ultimately, we obtain the query matrix $\boldsymbol{Q}_p$ (from the peptide sequence $\boldsymbol{X}_p$) and the key and value matrices $\boldsymbol{K}_M, \boldsymbol{V}_M$ (from the MHC sequence $\boldsymbol{X}_M$). A **cross-attention** module, as shown in Figure 2(d), then takes these three matrices as inputs and dynamically integrates the peptide and MHC sequences through

$$\text{Attn}(\boldsymbol{Q}_p, \boldsymbol{K}_M, \boldsymbol{V}_M) = \text{softmax}\Big(\boldsymbol{Q}_p \boldsymbol{K}_M^\mathsf{T}/\sqrt{d}\Big)\boldsymbol{V}_M,$$

The sequence matrix $\text{Attn}(\boldsymbol{Q}_p, \boldsymbol{K}_M, \boldsymbol{V}_M)$ (which shares the same shape as the peptide query matrix $\boldsymbol{Q}_p$) produced by the last cross-attention layer is then refined through normalization and feed-forward layers and undergoes mean pooling; ultimately, the resulting vector is fed into a classifier and trained with cross-entropy loss.

**Remark.** Through this stage, we obtain the intermediate model **Fusion-pM**, which effectively leverages the readily available peptide–MHC binding data that is far more abundant than peptide–MHC–TCR triad binding data. A similar practice was also adopted by Lu et al. (2021).

**Stage 2: Full-parameter fine-tuning for peptide–MHC–TCR binding.** We then train the whole model. Here is how we handle the three sequences: As shown in Figure 2(a), we pass peptide and MHC sequences to the pre-trained model from Stage 1, wherein the aforementioned mean-pooling module and the classifier are removed; the sequence matrix produced by the last cross-attention layer in the pre-trained peptide–MHC part is then transformed into a query matrix $\boldsymbol{Q}_{\text{pM}}$ and interacts with the TCR key and value matrices $\boldsymbol{K}_T, \boldsymbol{V}_T$ in a cross-attention module, yielding

$$\text{Attn}(\boldsymbol{Q}_{\text{pM}}, \boldsymbol{K}_T, \boldsymbol{V}_T) = \text{softmax}\Big(\boldsymbol{Q}_{\text{pM}} \boldsymbol{K}_T^\mathsf{T}/\sqrt{d}\Big)\boldsymbol{V}_T.$$

**Remark.** In this stage, the model is further trained on peptide–MHC–TCR binding data, resulting in the full **Fusion-pMT** model. In particular, we preserve the sequence form for both the TCR and the peptide–MHC interaction product as discussed in Section 4.1, and we apply the unified encoder consistently across all three sequences (Section 4.2).

This two-stage paradigm allows realistic modeling of interactions among the peptide, MHC, and TCR sequences. Initially, the peptide and MHC representations interact to produce an intermediate sequence matrix, which is then used to interact with the TCR representation, capturing the complex dependencies among these biological sequences.

### 4.4 Implementation details

For the architecture of Fusion-pMT, we set the embedding dimension to 64, use 4 attention heads. Here, we would like to further discuss the practical issues and the related implementation details crucial to the model performance, as a closing remark to this methodology section.

**Gradient Vanishing.** To mitigate the issue of gradient vanishing, we employ the LeakyReLU activation function (Jha et al., 2022) in both the intermediate layers and the feedforward layers. Additionally, we implemented residual connections that bypass the attention mechanism by directly connecting the encoded sequence information to the fully connected layers, which reduces the risk of gradient vanishing.

# 5    Experimental Results and Analysis

This section comprises three main parts. We first describe the experimental setup in Section 5.1, followed by the baseline models and evaluation benchmarks in Section 5.2. Then we present the results on two core tasks: Peptide–MHC Binding and Peptide–MHC–TCR Binding (respectively in Sections 5.3 and 5.4), which involve out-of-distribution (OOD) evaluation (detailed in Section 5.1) to assess generalization. Finally, we conduct ablation studies in Section 5.5 to examine the contributions of key model components, including the sequence representation and the unified encoder.

## 5.1    Experiment Setups

The experiments along this section are mainly conducted to examine the performance of two important variants featured with our proposed techniques: **Fusion-pM** (the model for predicting **peptide–MHC binding**) and **Fusion-pMT** (the model for predicting tripartite **peptide–MHC–TCR binding**). Overall, both of the models were implemented in PyTorch and conducted on an NVIDIA A100 40 GB. In training, we used a batch size of 64 and a learning rate of 0.01, an Adam optimizer, and trained the models for 300 epochs.

**Peptide–MHC binding.**    For the model training and testing, we align our testing protocols and datasets with the ones from the previous work TransPHLA (Chu et al., 2022). Specifically, TransPHLA organizes the data into multiple partitions: Training, Validation, Testing (called `Independent` in  (Chu et al., 2022)), where the testing set is used to test the model's generalization to unseen alleles, and designated as **pHLA Testing Dataset**.

**Peptide–MHC–TCR binding.**    For model training, validation and testing, we used a dataset derived from Lu et al. (2021), referred to as the **pMT training**, **pMT validation** and **pMT testing**, with a positive-to-negative sample ratio of 1:10. For details, we closely follow the experimental setup used in pMTnet (Lu et al., 2021) and pMTnet-omni (Han et al., 2023). We adhere to the original preprocessing protocols, including sequence amendments and negative pair generation, to ensure consistency and comparability with prior work, i.e., the positive samples were further augmented tenfold, resulting in an effective 1:1 ratio between positive and negative samples during training. Specifically, this dataset includes 28,604 unique TCR CDR3 sequences, 426 peptides, and 63 HLA types (MHC alleles). Aligning with pMTnet (Lu et al., 2021), we utilized the same **pMT testing dataset** from the GitHub of Lu et al. (2021), which contains 272 TCR sequences, 224 peptide sequences, and 24 MHC sequences, selected to ensure that the peptides are entirely unseen in the training or validation data. Overall, the dataset contains 32,607 pMHC–TCR positive pairs and a significantly larger set of generated negative pairs. Similarly to Lu et al. (2021), we then partitioned the training data into a training set and a validation set using an 80%:20% split to support model selection and hyperparameter tuning.

To further evaluate the model's out-of-distribution (OOD) generalization, we curated an additional **OOD testing dataset** using samples collected from VDJdb (Goncharov et al., 2022), which can be regarded as arising from a distinct sampling process compared to the original datasets (Lu et al., 2021). This OOD set consists of 1,346 TCR sequences, 239 peptide sequences, and 53 MHC sequences, and was designed to challenge the model's robustness when applied to data distributions not seen during training.

In both test datasets, negative samples were randomly generated (the negative samples in **pMT testing dataset** were copied down from the official GitHub of Lu et al. (2021)), and all positive samples previously included in the training or validation sets were explicitly excluded. This results in an imbalanced testing setup, maintaining a 1:10 positive-to-negative ratio, which reflects real-world biological conditions (Lu et al., 2021).

## 5.2    Baselines and Benchmarks

Despite the growing interest in computational immunology, the field still lacks standardized datasets and benchmark protocols for evaluating interactions among T cell receptors (TCRs), major histocompatibility complex (MHC) molecules, and antigenic peptides. This absence of widely accepted benchmarks hinders robust comparison, reproducibility, and scientific progress. Establishing reliable baselines and benchmarks

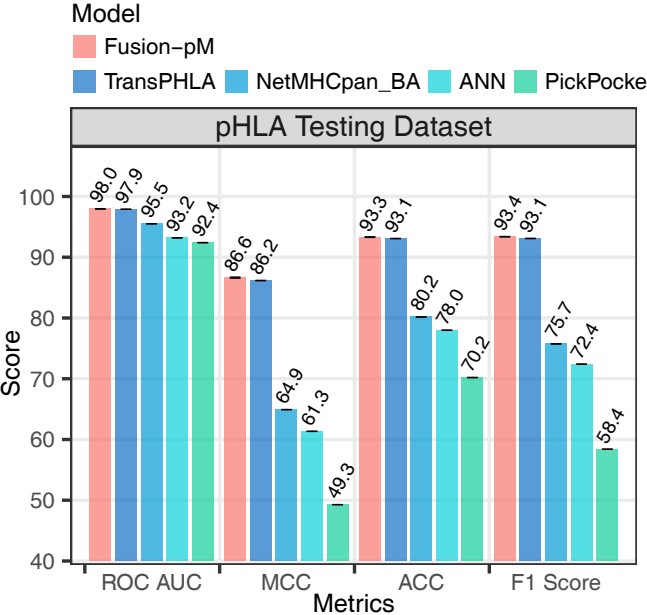

Figure 3: Comparison of peptide–MHC binding models based on multiple evaluation metrics (%), including ROC AUC, MCC, ACC, and F1 Score. The models compared are Fusion-pM, TransPHLA, NetMHCpan_BA, ANN, and PickPocket, evaluated on "pHLA Testing Dataset", the testing dataset of TransPHLA (Chu et al., 2022).

is thus crucial for advancing the predictive capabilities and practical impact of computational models in immunology.

For the peptide–MHC (pMHC) binding task, we compare against several strong baselines, including TransPHLA (Chu et al., 2022), netMHCpan (Borole & Rajan, 2024), ANN (Choi et al., 2011), Pick-Pocket (Zhang et al., 2009), CcBHLA (Wu et al., 2023), and STMHCpan (Ye et al., 2023). We evaluate model performance using multiple standard metrics, including Accuracy (ACC), F1 Score, Matthews Correlation Coefficient (MCC), and Receiver Operating Characteristic Area Under the Curve (ROC AUC).

For the peptide–MHC–TCR binding task, we closely follow the experimental setups defined in pMTnet (Lu et al., 2021) and pMTnet-omni (Han et al., 2023), using the same datasets, preprocessing protocols, and data splits as described in Section 5.1. In this context, we benchmark our models against four state-of-the-art baselines specifically designed for peptide–MHC–TCR binding prediction or general protein interaction modeling: pMTnet (Lu et al., 2021), PISTE (Feng et al., 2024), ESM-2 (Lin et al., 2023), and ERGO II (Montemurro et al., 2021). These baselines reflect diverse modeling approaches, including LSTM-based architectures, sliding and cross attention mechanisms, and large-scale pretrained protein language models. We evaluate model performance using multiple standard metrics, including Accuracy (ACC), F1 Score, Matthews Correlation Coefficient (MCC), Receiver Operating Characteristic Area Under the Curve (ROC AUC), and Precision–Recall Area Under the Curve (PR AUC).

In addition, we provide the complete evaluation results of our Fusion-pMT on the peptide–MHC–TCR binding task across four distinct sets—pMT training, validation, testing, and OOD testing sets—not only covering the five core metrics mentioned above but also including sensitivity, specificity, precision, and recall to offer a more comprehensive assessment of our model performance.

## 5.3 Immune Presentation Prediction (Peptide–MHC Binding)

As a byproduct of Stage 1 pre-training, our peptide–MHC binding model **Fusion-pM** provides a lightweight yet competitive alternative to existing methods. Unlike TransPHLA (Chu et al., 2022), which employs

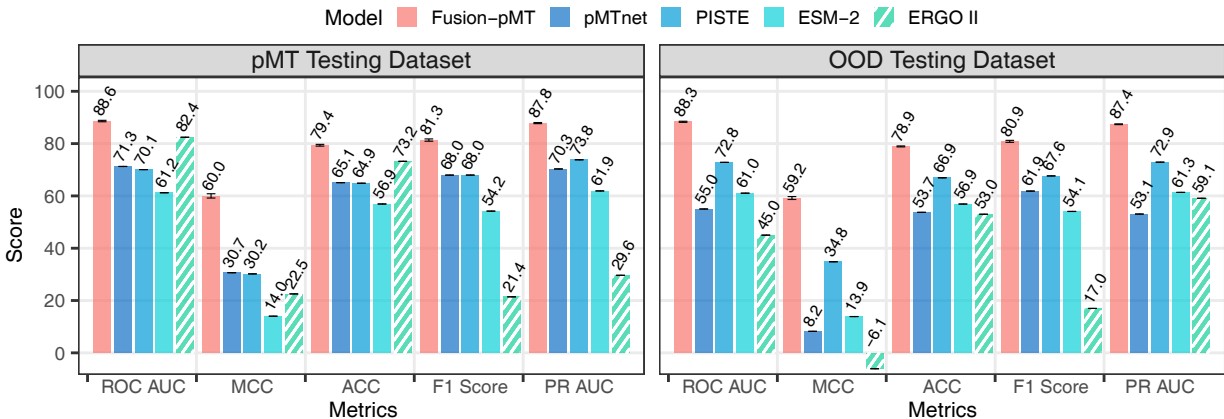

Figure 4: Performance (%) of the five tri-molecular binding prediction models on pMT and OOD testing datasets in terms of ROC AUC, ACC and MCC. Here, ERGO II was developed and as well evaluated solely for peptide-TCR binding prediction, serving as a sanity check that incorporating more molecules is beneficial; other methods were tested on peptide–MHC–TCR triad binding prediction. "pMT Testing Dataset" is the original testing dataset in Lu et al. (2021). "OOD Testing Dataset" is a new dataset collected from Goncharov et al. (2022). Notably, the calculation of the metric "MCC" hinges on correlation, which thus can be negative.

Table 2: Performance (%) Metrics of Fusion-pMT Model: Mean and Standard Deviation of ROC AUC, ACC, MCC, F1 Score, Sensitivity, Specificity, Precision, Recall, and PR AUC values on pMT validation, training, Testing and OOD Datasets.

| Datasets | Metrics Score (mean $\pm$ std) | | | | | | | | |
|---|---|---|---|---|---|---|---|---|---|
| | ROC AUC | ACC | MCC | F1 Score | Sensitivity | Specificity | Precision | Recall | PR AUC |
| pMT Validation | $88.36_{\pm 0.25}$ | $79.04_{\pm 0.35}$ | $59.48_{\pm 0.84}$ | $80.96_{\pm 0.43}$ | $89.39_{\pm 2.43}$ | $68.75_{\pm 2.68}$ | $74.04_{\pm 1.16}$ | $89.39_{\pm 2.43}$ | $87.25_{\pm 0.19}$ |
| pMT Training | $92.70_{\pm 0.23}$ | $83.53_{\pm 0.41}$ | $68.01_{\pm 0.61}$ | $84.73_{\pm 0.27}$ | $91.43_{\pm 2.25}$ | $75.64_{\pm 2.86}$ | $79.02_{\pm 1.56}$ | $91.43_{\pm 2.25}$ | $92.40_{\pm 0.20}$ |
| pMT Testing | $88.63_{\pm 0.27}$ | $79.38_{\pm 0.34}$ | $60.01_{\pm 0.82}$ | $81.35_{\pm 0.43}$ | $89.45_{\pm 2.51}$ | $69.18_{\pm 2.79}$ | $74.66_{\pm 1.21}$ | $89.45_{\pm 2.51}$ | $87.85_{\pm 0.21}$ |
| OOD Testing | $88.35_{\pm 0.21}$ | $78.93_{\pm 0.18}$ | $59.19_{\pm 0.55}$ | $80.86_{\pm 0.38}$ | $89.05_{\pm 2.75}$ | $68.81_{\pm 3.04}$ | $74.13_{\pm 1.28}$ | $89.05_{\pm 2.75}$ | $87.40_{\pm 0.15}$ |

multi-layer self-attention with millions of parameters, Fusion-pM integrates two compact architectural components—a *Unified Encoder* for peptides and MHCs, and a *Cross-Attention Sequence Fusion Block*. This design enables consistent feature extraction across heterogeneous inputs and explicitly models dynamic peptide–MHC interactions, all within a parameter budget of fewer than 700k parameters.

Despite being significantly smaller, Fusion-pM achieves stable and consistently strong results across all four metrics. As shown in Figure 3, it slightly surpasses TransPHLA in accuracy (93.30% vs. 93.08%) and F1 score (93.40% vs. 93.10%), while also yielding higher MCC (86.60 vs. 86.20) and ROC AUC (98.00 vs. 97.90). These results highlight that Fusion-pM not only maintains predictive performance comparable to state-of-the-art baselines, but also achieves superior efficiency in terms of parameter usage and inference speed—making it highly practical for large-scale or resource-constrained biological applications.

### 5.4 Immunogenicity Prediction (Peptide-MHC-TCR Binding)

**pMT testing dataset.** Figure 4 presents results on the pMT test set across five peptide–MHC–TCR triad models. Fusion-pMT attains a ROC AUC of 88.63% and a PR AUC of 87.85%, notably surpassing pMTnet (71.28% and 70.30%), reflecting a stronger ability to capture peptide–MHC–TCR binding relationships. Its MCC reaches 60.0% versus 30.7% for pMTnet. Because MCC balances TP, TN, FP, and FN and is robust to class imbalance, this indicates Fusion-pMT learns meaningful immunogenicity patterns rather than overfitting or relying on biased decision rules, and generalizes across both positive and negative classes.

Although ERGO II achieves relatively high ROC AUC and ACC (82.4%/73.2%), its MCC/F1/PR AUC (14.2%/24.1%/26.6%) reveal strong negative-class bias under the ∼90% negative prevalence: high ACC/ROC

Table 3: Statistical Comparison of Fusion-pMT Models: T-value and P-value for ACC, ROC AUC, and PR AUC, on Peptide-MHC-TCR binding task. [N] indicates the model is equipped with the fusion technique from netMHCpan (Jurtz et al., 2017).

| Metrics | Fusion-pMT[N] vs. pMTnet | | Fusion-pMT vs Fusion-pMT[N] | |
|---|---|---|---|---|
| | $t$-value | $p$-value | $t$-value | $p$-value |
| PR AUC | 55.00 | 0.000330 | 57.74 | 0.000001 |
| ROC AUC | 25.69 | 0.001512 | 10.32 | 0.008406 |
| ACC | 42.21 | 0.000561 | 85.98 | 0.000001 |

AUC can be attained by largely ranking and predicting negatives, while failing to capture true positive binding events. In contrast, Fusion-pMT delivers balanced performance across all metrics, indicating a deeper grasp of the underlying interactions.

To provide a fuller assessment, we also report sensitivity, specificity, precision, and recall in Table 2. Fusion-pMT achieves 89.45% sensitivity (true positive rate) and 69.18% specificity (true negative rate), offering a nuanced view of both detecting binders and rejecting non-binders, and confirming robust, consistent performance.

**Out-of-distribution testing dataset.** Figure 4 summarizes OOD performance across five peptide–MHC–TCR triad models. Fusion-pMT consistently surpasses pMTnet: on the pMT test set, ROC AUC 88.36% vs 78.91%, ACC 78.9% vs 53.7%, with pMTnet's MCC only 8.2 (near chance). Fusion-pMT shows comparable performance on the OOD set from Goncharov et al. (2022) and the in-distribution pMT test from Lu et al. (2021), indicating it learns biologically meaningful features rather than memorizing. With fewer than 700k parameters, it is compact and efficient. Statistical comparisons in Table 3 confirm significant gains across ACC, ROC AUC, and PR AUC (all $t > 25$, $p < 0.002$), ruling out random fluctuations. Performance on pMT testing dataset and OOD testing Dataset is a critical indicator of a model's potential clinical applicability (Gao et al., 2023).

**Remark.** Overall, Fusion-pMT achieves consistent and significant improvements over all four baselines on both in-distribution and out-of-distribution testing datasets, demonstrating its strong generalization ability. At the same time, its byproduct model Fusion-pM attains comparable results to state-of-the-art methods on the peptide–MHC binding task with far fewer parameters. Together, these findings provide strong empirical evidence for the effectiveness of our two key architectural components: the *Unified Encoder* and the *Cross-Attention Sequence Fusion Block*.

## 5.5 Discussion: Sequence Representation Analysis

Compared with pMTnet using a bottleneck autoencoder model to encode the TCR sequence, we propose a cross-attention-based transformer, Fusion-pMT (netMHCpan), to represent the TCR sequence and preserve the sequence form until the binding prediction block (thus the pre-trained module netMHCpan is kept, as in pMTnet, for fair comparison). The model architecture is in Section 4. Fusion-pMT (netMHCpan) shows significant improvements in performance metrics over pMTnet, as evidenced from both numerical comparisons (e.g., ACC, PR AUC, and ROC AUC in Figure 5) and statistical significance tests (e.g., $t$-tests in Table 3). These considerable improvements demonstrate the critical importance of maintaining sequence integrity in our model, which enables more effective capturing of complex, sequence-dependent interactions crucial for accurate binding predictions.

## 5.6 Ablation studies: unified encoders

To ablate the usage of unified encoders, we introduce a model variant, Fusion-pM (w/o unified), which employs *distinct* encoders for peptides and MHC sequences and similarly incorporates a cross-attention

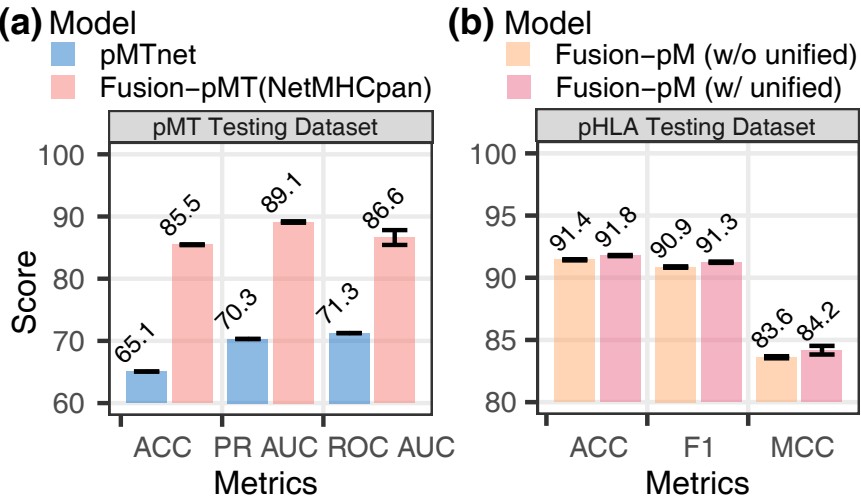

Figure 5: Ablation studies for **(a)** Peptide-MHC-TCR triad binding prediction with sequence representation and **(b)** peptide–MHC models with unified encoders. Notably, "pMT / pHLA Testing Dataset" is the original testing dataset in Lu et al. (2021) / Chu et al. (2022).

sequence fusion block. This specification allows a fair comparison with the base Fusion-pM (w/ unified) variant, which we recall employs a unified encoder for both peptides and MHCs.

As shown in Figure 5(b), the results demonstrate that Fusion-pM (w/ unified) statistically significant improvements across all metrics. The ACC scores exhibit minimal (though significant) variation between the two model variants; however, both F1 and MCC metrics indicate more substantial gains with the Fusion-pM (w/ unified) configuration, suggesting that employing the unified encoder for sequence fusion not only simplifies the model architecture but may also enhance performance in terms of both prediction precision and class balance handling (Cer et al., 2018). These results prompt further investigation into the benefits of encoder uniformity in complex sequence fusion tasks in immunological prediction.

## 6 Conclusions

In this paper, we have proposed a new model **Fusion-pMT** for biological language modeling in peptide–MHC–TCR triad binding, through a revisit of sequence fusion mechanisms in representation learning research. A key insight is that *maintaining the sequence form throughout the transformation* not only aligns with the real biological processes but also significantly improves immunogenicity prediction. Building on this, we characterize interactions across different sequences through both a cross-attention mechanism and a unified amino acid embedding vocabulary. This unified encoder design enables parameter sharing across heterogeneous inputs, requiring fewer than 700k parameters in total. By avoiding multi-layer self-attention stacks, Fusion-pMT remains lightweight and effective. Moreover, following the two-stage biological binding pathway (1. peptides first bind MHCs and 2. the resulting complexes are then recognized by TCRs) allows us to naturally obtain **Fusion-pM** in Stage 1, which serves as a strong peptide–MHC predictor.

Our extensive experiments further corroborate that Fusion-pMT consistently outperforms strong baselines across both in-distribution and out-of-distribution datasets, while Fusion-pM achieves results comparable to or even surpassing state-of-the-art peptide–MHC binding models. These results highlight that our framework attains strong performance while remaining highly efficient, making it well-suited for potential applications to diverse and dynamic biological scenarios. Overall, our work presents a biologically motivated and computationally efficient framework for modeling peptide–MHC–TCR interactions, offering valuable insights and a foundation for future models that aim to integrate more complex and heterogeneous biological languages.

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

# A  Protein Sequence Embedding

In this appendix, we introduce the tedious notations for the concrete protein sequence embedding in our models. Our approach is designed to preserve the intrinsic sequential integrity of these sequences throughout the encoding and transformation processes; these details are vital for understanding the complex interactions within biological sequences.

## A.1  Protein Sequences

We start with the embedding of protein sequences in this subsection, and then introduce the positional encoding method we adopt in Appendix A.2.

A protein sequence $S$ is composed of $L$ amino acids, which can be mathematically represented as

$$S = [a_1, a_2, a_3, \ldots, a_L].$$

Here, each $a_i$ indicates a certain amino acid, and the collection of the 21 standard amino acids is denoted as

$$\mathcal{A} := \{\texttt{A}, \texttt{C}, \texttt{D}, \texttt{E}, \texttt{F}, \texttt{G}, \texttt{H}, \texttt{I}, \texttt{K}, \texttt{L}, \texttt{M}, \texttt{N}, \texttt{P}, \texttt{Q}, \texttt{R}, \texttt{S}, \texttt{T}, \texttt{V}, \texttt{W}, \texttt{X}, \texttt{Y}\}$$

in terms of letter abbreviations of amino acids.

**One-hot encoding.**  Following Devlin et al. (2019), we first transform a protein sequence into a binary vector representation, which is a common practice in the representation of textual sequences. Here, each amino acid $a_i$ corresponds to a one-hot encoded vector $\boldsymbol{h}(a_i)$ of length $|\mathcal{A}|$:

$$\boldsymbol{h}(a_i) = [h_1, h_2, \ldots, h_{|\mathcal{A}|}]^T,$$

where $h_j = 1$ if $a_i$ is the $j$-th amino acid in $\mathcal{A}$, and $h_j = 0$ otherwise. The entire protein sequence $S$ is thus

$$\mathbf{H}^T(S) = [\mathbf{h}(a_1), \mathbf{h}(a_2), \ldots, \mathbf{h}(a_L)],$$

where the sequence matrix $\mathbf{H}(S) \in \{0, 1\}^{L \times |\mathcal{A}|}$.

**Encoding**  Notably, each amino acid $a_i$ can be encoded using another specific numerical representation that captures its chemical properties and contributes to its role within the protein structure. This encoding might utilize techniques ranging from simple categorical encoding schemes to more complex embeddings derived from machine learning models.

**Transformation processes**  The encoded representations are processed through computational models (e.g., convolutional neural networks or recurrent neural networks) designed to capture the interactions between amino acids and to preserve their positional information.

**Aggregation**  The transformed representations are ultimately aggregated to form a comprehensive vector representation of the entire sequence. This step may involve methods like pooling.

Overall, these steps, which are well-studied in representation learning literature, ensure that our model not only captures the individual characteristics of each amino acid but also their contextual relationships within the entire sequence.

## A.2  Positional Encoding for Protein Sequences

The positional encoding provides the model with information about the relative or absolute position of the tokens in the sequence. Let $s$ denote the position within the biological sequence, $i$ be the dimension within the embedding spaces, and $d$ indicate the dimensionality of the model embeddings. One can then apply the

sine and cosine functions for positional encoding as follows Vaswani et al. (2017):

$$p(s, 2i) = \sin\left(\frac{s}{10000^{2i/d}}\right),$$
$$p(s, 2i + 1) = \cos\left(\frac{s}{10000^{2i/d}}\right),$$

The encoding mechanism ensures that the model can effectively interpret the sequential order of the sequences for biological interactions, and we apply this positional encoding in computing the cross-attention modules.

## B  Useful Facts

### B.1  Biological Molecules of Adaptive Immunity

In adaptive immunity, the major players are the highly diverse B and T cells, with unique surface receptors known as B cell receptors (BCRs) and T cell receptors (TCRs), respectively. These cells recognize specific parts of an antigen, referred to as epitopes. However, the mechanisms of antigen recognition differ between B and T cells. B cells target a fragment of the antigen known as a B cell epitope. Recognition by BCRs primarily depends on three-dimensional conformational information from the fragment, which contains mainly non-contiguous amino acid residues. On the other hand, T cell epitopes, recognized by TCRs, depend on their binding to major histocompatibility complex (MHC) molecules. These epitopes are linear, formed by contiguous amino acid residues.

**Major Histocompatibility Complex (MHC)**  The **major histocompatibility complex (MHC)** is a type of cell surface proteins essential for the adaptive immunity. In humans, MHC genes are called human leukocyte antigens (*HLA*s). The MHC class I molecules present endogenous peptides from proteins self-generated intracellularly, while The MHC class II molecules are mainly expressed on antigen presenting cells. The MHC class I molecules contain an $\alpha$ chain from *MHC* class I genes and $\beta_2$ microglobulin ($\beta_2 m$), which can present peptides ranging from 8 to 12 amino acids. MHC class II molecules consist of one $\alpha$ and one $\beta$ chain, allowing the binding of longer peptides ranging from 9 to 25 amino acid residues, or even longer. The MHC class I and MHC class II are also highly diverse, with approximately $6 \times 20^{6-7}$ and $12 \times 20^{10}$ alleles, respectively (Rock et al., 2016). The MHC class I molecules present endogenous peptides from proteins self-generated intracellularly, while The MHC class II molecules are mainly expressed on antigen presenting cells.

**T Cell Receptor (TCR)**  The **T cell receptor (TCR)** is a type of protein complex on the surface of T cells responsible for recognizing fragments of antigen as peptides bound to MHCs. Classically, the TCR consists of an $\alpha$ chain and a $\beta$ chain, which are encoded by gene *TRA* and *TRB*, respectively. The high diversity of TCR is generated by rearrangements of the V and J segments of the *TRA* gene and V, D, and J segments of the *TRB* gene in the thymus, with $10^{23}$ possible rearrangements theoretically (Mora & Walczak, 2019). Within the TCRs, the indices for $\alpha$ and $\beta$ chains have been separately estimated to be $10^9$ and $10^{14}$ (Mora & Walczak, 2019). Consequently, $\beta$ chains garner a greater degree of attention and are the focus of significant experiments in TCR sequencing, making $\beta$ chains a core component in data-driven modeling. In another estimation, the number of potential rearrangements can be up to $10^{61}$ (Chi et al., 2024). While at one moment, there are around $10^{11}$ per human with around $10^9$ distinct TCRs (Chi et al., 2024), which requires the highly precise prediction of pMHC–TCR for further drug development based on TCRs.

**B Cell Receptor (BCR)**  The **B cell receptor (BCR)** from B cells contains multiple forms, including the secreted form and the membrane-bound form. Secreted BCRs are usually called **antibody** (**Ab**), while both membrane-bound and secreted BCRs can be called **immunoglobulin** (**Ig**). BCRs are arranged in three globular regions that roughly form a Y shape. In humans, one BCR unit consists of four chains, two heavy chains (H) and two light chains (L). Each heavy chain's variable region is approximately 110 amino acids in length. There are five types of mammalian BCR heavy chains denoted by Greek letters: $\alpha$, $\delta$, $\varepsilon$, $\gamma$ and $\mu$. These chains are found in **IgA**, **IgD**, **IgE**, **IgG**, and **IgM** antibodies, respectively. Heavy chains differ in size and composition. $\alpha$ and $\gamma$ contain approximately 450 amino acids, while $\varepsilon$ and $\mu$ have about 550 amino acids.

Table 4: Overview of Molecules Involved in Antigen Presentation.

| Molecule | Location | Total Length (aa) | Active Region | Main Function | Theoretic Diversity | Homology |
|---|---|---|---|---|---|---|
| MHC Class I | Cell Surface | $\alpha$: $\sim$360 $\beta_2 m$: $\sim$120 | Relevant: $\alpha$ chain | Present peptides to $CD8^+$ T cells | $6 \times 20^{6-7}$ | Varies |
| MHC Class II | Cell Surface | $\alpha$ & $\beta$: 260–280 | Relevant: $\alpha_1$ and $\beta_1$ domains | Present peptides to $CD4^+$ T cells | $12 \times 20^{10}$ | Varies |
| T Cell Receptor | T Cell Surface | $\alpha$: 223 $\beta$: 247 | Variable regions: 110–120 each chain | Recognize peptide–MHC complexes | $10^{23}$ | Low |
| B Cell Receptor | B Cell Surface or Secreted Form | Light: 211–217 Heavy: $\sim$450/$\sim$550 | Variable domain: 110 | Recognize antigens | $10^{21}$ | Low |

In mammals, there are only two types of light chains, $\lambda$ and $\kappa$, which have minor differences in the sequence. A light chain has two successive domains, constant ($C_L$) and variable ($V_L$). The approximate length of a light chain is 211–217 amino acids. The diversity of BCR is generated from V(D)J recombination and somatic hypermutation, with $10^{21}$ possible rearrangements theoretically (Mora & Walczak, 2019). Another estimation suggested that the total paired-sequence diversity is $10^{16-18}$, while there are $5 \times 10^9$ B cells in the peripheral blood of a healthy human.

### B.2  Amino Acids Embedding

Accurately embedding amino acids is key to modeling protein interactions, as it preserves each amino acid residue's biochemical properties and positional context. Approaches include position-specific scoring matrices (Madrigal et al., 2024) and deep learning-based embeddings (Cao et al., 2021; Tu et al., 2022; Lee et al., 2021), which help maintain the structural and functional integrity of the sequence. Advanced embeddings often integrate attention mechanisms to capture long-range dependencies among amino acid residues, thereby improving representation of spatial relationships crucial for protein functionality (Reynisson et al., 2020).

### B.3  Protein Interactions and Prediction Methods

In our computational study, we developed a specialized neural network model, termed Fusion-pMT, to understand the interactions within the peptide–MHC–TCR complex. The model's architecture leverages a custom-built submodule, which employs an advanced multi-head attention mechanism (with eight attention heads and a dropout rate of 0.1) to process and integrate features from peptide and MHC sequences. The sequences are embedded into a 64-dimensional space, facilitating a detailed representation of their complex biological characteristics.

The model encapsulates the dynamics of peptide–MHC interactions through its cross-attention mechanism, which is crucial for capturing the nuanced dependencies between these biomolecules. Further processing is performed by a fully connected neural network, which integrates the attention outputs with flattened peptide and MHC sequence features. This integration feeds into a deep learning pipeline that includes multiple layers of nonlinear transformations and dropout regularization, aiming to predict interaction outcomes robustly.

Training of the pMHC Model is meticulously orchestrated over 200 epochs, employing a binary cross-entropy loss function optimized via stochastic gradient descent with a learning rate of 0.1. This training regimen includes a patience mechanism set to 10 epochs to prevent overfitting and ensure model generalizability. Model performance is evaluated through both training and validation phases, with checkpoints saved upon achieving new best validation accuracies, underscoring the model's progressive learning capability.

## C  Training Details

The integration model processes combined features through a carefully designed series of fully connected layers, reducing the dimensionality from the combined inputs to a singular output that signifies the likelihood of interaction. To mitigate the risk of overfitting, given the model's complexity and the intricate nature of the immunological data, dropout layers with a rate of 0.1 are included following each activation phase. Following the implementation details in the main text, we employ LeakyReLU activations in intermediate and

feed-forward layers and use residual connections to alleviate gradient vanishing. The transformer components use an embedding dimension of 64 and 4 attention heads.

All models are implemented in PyTorch and trained on NVIDIA A100 40 GB GPUs. Unless otherwise noted, models are trained for 300 epochs with a batch size of 64. We use the Adam optimizer with a learning rate of 0.01 and the Binary Cross-Entropy with Logits loss function (nn.BCEWithLogitsLoss()). The validation split is used for model selection and hyperparameter tuning as described in the main text.

Despite occasional fluctuations in validation loss, the combination of dropout regularization, residual connections, and the compact parameterization contributed to stable training dynamics. The resulting models exhibit strong generalization, including on out-of-distribution evaluations reported in the main text.

**Avoiding Overfitting.** We primarily rely on dropout (rate 0.1), residual connections, and a compact parameter budget to reduce overfitting risk. Model selection is conducted on the validation set. No early stopping is used in the main experiments unless explicitly stated.

Table 5: Hyperparameter settings for the Peptide–MHC–TCR interaction model

| Parameter | Value | Description |
| --- | --- | --- |
| Embedding Dimension | 64 | Dimensionality of sequence embeddings |
| Attention Heads | 4 | Number of heads in the multi-head attention layers |
| Optimizer | Adam | Optimization algorithm |
| Learning Rate | 0.01 | Step size at each iteration of model weights |
| Batch Size | 64 | Number of samples per batch |
| Dropout Rate | 0.1 | Proportion of neurons disabled during training |
| Training Epochs | 300 | Number of complete passes through the training dataset |
| Loss Function | BCEWithLogits | Binary Cross-Entropy with Logits loss |

## D  Miscellanies

### D.1  Impacts on immunology and medicine

By using cross attention to address multi-sequence biological problems, the prediction of pMHC–TCR has implications in various fields. In immunology, an AI4Sci model understanding the TCR-pMHC interaction can help in the study of diseases, including autoimmune diseases, infections, and cancers. In medicine, AI-empowered predictions of TCR-pMHC interactions can potentially lead to individualized treatments in precision medicine.

### D.2  Broader Impact and Limitations

While Fusion-pMT shows promise, we acknowledge reliance on databases like IEDB/VDJdb, which historically overrepresent European-descent HLA alleles. This data bias may affect predictive accuracy for underrepresented populations. Clinical application requires rigorous experimental validation to prevent potential health disparities.

### D.3  Impacts in Healthcare

In healthcare, the prediction of pMHC–TCR interactions has significant implications in the development of advanced therapies against cancers or infectious diseases. (Figure 6) .

**Vaccine Development.** Understanding which peptides can bind to MHC molecules and be recognized by TCRs can help in the design of more effective vaccines, especially the neoantigen-based cancer vaccine.

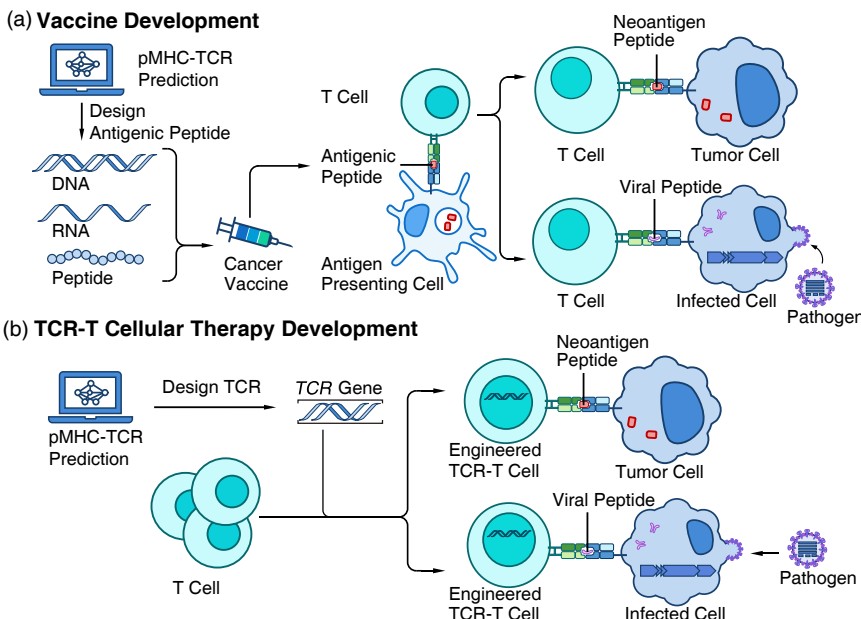

Figure 6: **The Application of pMHC–TCR Binding Prediction in Healthcare (a)** Schematic Representation of the Therapeutic Cancer Vaccine. **(b)** Schematic Representation of the Engineered TCR-T Cell Therapy.

Neoantigens are newly generated peptides from somatic mutations that can be recognized by TCRs of tumor-specific T cells. Once all mutations are identified, they must be computationally predicted from matched tumor-normal sequencing data, and then ranked according to their predicted capability in stimulating a T cell response. Neoantigen-based cancer has shown promising results in a phase IIb study (Weber et al., 2024). This selection of effective neoantigen candidates relies on the precise prediction of pMHC–TCR interactions (Rojas et al., 2023; Yarchoan et al., 2024).

In addition to cancers, pMHC–TCR prediction can also accelerate the development of infectious disease vaccines. During the COVID-19 pandemic, T-cell-directed vaccines has been designed in the form of peptides (Heitmann et al., 2022) and mRNA (Arieta et al., 2023). More precise prediction of pMHC–TCR interactions can improve the development of T-cell-directed vaccines in patients with immunodeficiency in phase I/II study (Heitmann et al., 2023) (Figure 6(a)).

**TCR-T Cellular Therapy.** The prediction of pMHC–TCR interactions can also aid in the development of T cell therapies, where the goal is to enhance the immune system's ability to recognize and destroy abnormal cells. The development of TCR-T therapies against cancer involves identifying a specific TCR that recognizes the tumor antigen by analyzing TCR sequencing data. Subsequently, this *TCR* gene can be manipulated to be expressed in autologous T cells. These engineered tumor-specific T cells can be expanded to induce tumor killing by recognizing pMHC on tumor cells (Hassel et al., 2023; D'Angelo et al., 2024). Two drug based on this therapy has been approve by the Food and Drug Administration of USA on January 25, 2022 (Mullard, 2022) and August 2, 2024, respectively. Additionally, the proof-of-concept for using TCR-T therapies against infectious diseases have been validated in treating cytomegalovirus infection after hematopoietic stem cell transplantation (Liu et al., 2022), which sheds light on the broader application of TCR-T cellular therapies(Figure 6(b)).

### D.4  Asset Licenses

- TransPHLA-AOMP (Chu et al., 2022): TransPHLA-AOMP (transformer-based model for pHLA binding prediction and the automatically optimized mutated peptides program) is an algorithm designed to predict peptide and HLA binding. TransPHLA-AOMP is licensed under the GNU GENERAL PUBLIC license 3.0.
- pMTnet (Lu et al., 2021): pMTnet (the pMHC–TCR binding prediction network) is an algorithm to predict TCR binding specificities of the neoantigens and T cell antigens in general presented by class I major histocompatibility complexes. pMTnet is licensed under the GNU GENERAL PUBLIC license 2.0.
- NetMHCpan (Reynisson et al., 2020): NetMHCpan (pan-specific binding of peptides to MHC class I proteins of known sequence) is an algorithm to predict the binding of peptides to any MHC molecule of a known sequence using artificial neural networks. NetMHCpan is licensed under the GNU GENERAL PUBLIC license 3.0.
- VDJdb (Goncharov et al., 2022): VDJdb is a curated database of T-cell receptor sequences of known antigen specificity. This database is licensed under the Attribution-NoDerivatives 4.0 International.

# E  Additional Experimental Analysis

In this appendix, we provide supplementary experiments and analyses to further validate the efficiency, robustness, and design choices of Fusion-pMT. These results address specific inquiries regarding model parameterization, out-of-distribution (OOD) validity, and the impact of the pre-training stage.

### E.1  Model Efficiency Analysis

To validate the "lightweight" design claim of Fusion-pMT, we conducted a direct parameter count comparison against state-of-the-art baselines, including domain-specific models (PISTE, pMTnet) and general protein language models (ESM-2).

As presented in Table 6, Fusion-pMT utilizes a compact architecture with fewer than 0.7 million parameters. In stark contrast, even the smallest version of ESM-2 requires approximately 8 million parameters, while the standard version reaches 650 million. Our model achieves competitive predictive performance with approximately **0.1%** of the parameters required by large PLMs. This high parameter efficiency makes Fusion-pMT significantly more suitable for high-throughput screening scenarios where inference speed and memory constraints are critical factors.

Table 6: Parameter Efficiency Comparison. Fusion-pMT achieves state-of-the-art performance with significantly fewer parameters compared to large Protein Language Models (PLMs) and other baselines.

| Model | Type | Backbone Architecture | Parameters (Approx.) | Relative Size |
|---|---|---|---|---|
| ESM-2 (Standard) | PLM | Transformer (33 layers) | ∼650 M | 928x |
| ESM-2 (Smallest) | PLM | Transformer (6 layers) | ∼8 M | 11.4x |
| PISTE | Domain-Specific | Sliding Attention | ∼4.5 M | 6.4x |
| pMTnet | Domain-Specific | VAE-CNN-ANN | ∼1.5 M | 2.1x |
| **Fusion-pMT (Ours)** | **Domain-Specific** | **Cross Attention Transformer** | **< 0.7 M** | **1x** |

### E.2  Out-of-Distribution Validity

To ensure that the high performance observed on the OOD Testing Dataset stems from the model's generalization capability rather than data leakage or similarity, we performed a t-SNE analysis on the feature representations of the peptides.

Figure 7 visualizes the distribution of peptide sequences from the Training set (Blue) and the OOD Testing set (Orange). The visualization reveals a clear distributional shift between the two datasets, confirming that the OOD dataset occupies a distinct region in the feature space. Despite this shift, Fusion-pMT maintains high predictive accuracy, demonstrating its robustness in learning generalized interaction rules rather than memorizing training samples.

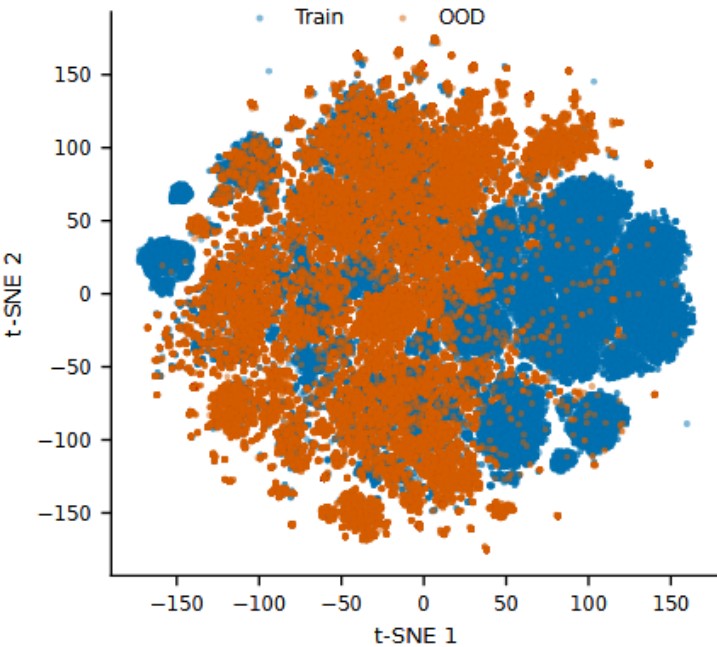

Figure 7: t-SNE visualization of peptide representations. The clear separation between Training samples (Blue) and OOD samples (Orange) confirms the validity of the out-of-distribution evaluation.

### E.3 Impact of Pre-training (Ablation Study)

To assess the necessity of the two-stage framework and the specific contribution of the pre-training stage (Fusion-pM), we conducted an ablation study by training a variant of Fusion-pMT from scratch (random initialization) without the Stage 1 constraint.

The results on the OOD dataset are summarized in Table 7. We observe that removing the pre-training stage leads to a performance drop in ROC AUC. This suggests that explicitly modeling the peptide-MHC binding event (Stage 1) acts as a crucial regularizer. It helps the model converge to a more biologically plausible minimum, thereby enhancing stability and generalization to unseen data. While the unified encoder and cross-attention mechanism provide the structural foundation, the pre-training strategy ensures the model respects the hierarchical nature of immune recognition.

Table 7: Ablation study on the OOD dataset. The "Drop" column indicates the decrease in ROC AUC compared to the full Fusion-pMT model.

| Model Variant | Description | ROC AUC Drop |
|---|---|---|
| **Fusion-pMT (Full)** | Proposed model with full two-stage training | - |
| w/o Pretraining | Trained from scratch (random init.) | -0.21% |
| w/o Shared Encoder | Separate encoders for Peptide, MHC, TCR | -0.40% |
| w/o Cross-Attention | Replaced with concatenation | -1.52% |
| w/o Bio-Workflow | Simultaneous processing (no stages) | -0.26% |

### E.4 Comparison with Pre-trained Protein Language Models

To further validate the design choices of Fusion-pMT, we address the potential alternative of utilizing large pre-trained protein language models (PLMs), such as ESM-2 (Lin et al., 2023). While replacing our domain-

specific encoder with a PLM might seem intuitive, our analysis and additional experiments demonstrate that simply fine-tuning ESM-2 is not an optimal solution for immunogenicity prediction.

We conducted comprehensive experiments using ESM-2 (8M and 35M parameters) as the backbone encoder under various settings. The results on the OOD Testing Dataset are summarized in Table 8. Our analysis reveals that ESM-2 lacks two critical design choices required for this specific task:

- **Lack of Native Interaction Mechanism:** ESM-2 is architecturally designed as a single-sequence Masked Language Model (MLM). It lacks the inductive bias to model the tripartite interaction between distinct molecules. As shown in Table 8, simply concatenating the embeddings from a frozen ESM-2 model ("Concat") yields suboptimal performance ($\approx 66.5\%$ ACC), which is significantly outperformed by Fusion-pMT (78.90% ACC). This confirms that the *Early Interaction* mechanism in our framework is essential.

Table 8: Performance comparison between Fusion-pMT and ESM-2 variants on the OOD Testing Dataset. "Bio-Workflow" denotes equipping the backbone with our proposed 2-stage Cross-Attention framework.

| Backbone Encoder | Fusion Strategy | Training Mode | ACC |
|---|---|---|---|
| **Fusion-pMT (Ours)** | **Bio-Workflow** | **Train from Scratch** | **0.7890** |
| ESM-2 (8M) | Concat | Frozen | 0.6650 |
| ESM-2 (35M) | Concat | Frozen | 0.6654 |
| ESM-2 (8M) | Concat | Fine-tune | 0.6723 |
| ESM-2 (35M) | Concat | Fine-tune | 0.7194 |
| ESM-2 (8M) | Bio-Workflow | Frozen | 0.8105 |
| ESM-2 (8M) | Bio-Workflow | Fine-tune | 0.6170 |

- Fine-tuning ESM-2 introduces significant computational overhead (increasing parameter count from $<0.7M$ to $>8M$ or $>35M$) without guaranteeing performance gains. The marginal gain observed with the frozen ESM-2 backbone equipped with our Bio-Workflow (0.8105 vs 0.7890) comes at the cost of massive model size, rendering it less suitable for high-throughput screening applications. Therefore, our domain-specific architecture, designed to learn binding physics from scratch, represents a more efficient and effective solution for this task.

