# OpenReview forum: "Fusion-pMT: Biological Language Modeling for Tri-Molecular Binding in Immunogenicity Prediction"
_TMLR — Rejected by TMLR_

### Review · Reviewer_D6kG · 2025-10-18

**Summary Of Contributions:**

The paper introduces Fusion-pMT, a biological language modeling framework designed to model the tri-molecular binding among peptide, MHC, and TCR, which is a central yet underexplored problem in immunogenicity prediction. The key motivation is that existing computational methods only address pairwise bindings (peptide–MHC or peptide–TCR), which fail to capture the full biological sequence of immune recognition. Fusion-pMT bridges this gap by translating biological interactions into a sequence-fusion task inspired by multimodal large language models.

Strengths:
1. Biological alignment: The model design explicitly mirrors the natural peptide–MHC–TCR binding cascade, improving interpretability and faithfulness to immunological processes.
2. Compact and efficient: Achieves state-of-the-art accuracy with an order-of-magnitude fewer parameters than baselines, making it suitable for biomedical deployment.
3. Comprehensive evaluation: Includes both in-distribution and out-of-distribution tests, along with thorough ablation and statistical analyses.

Weaknesses:
1. Weak novelty: The task itself is kind of novel, however, there is little technical novelty. The use of cross-attention and shared encoders, though biologically motivated, are incremental adaptations of standard transformer components.
2. Proof of interpretability: The authors claim that their method is motivated by improving interpretability, but there isn't any analysis or experiments about it.

**Audience:**

Yes

**Audience Explanation:**

The task about Peptide–MHC–TCR binding is kind of new and challenging. The paper is well-presented and easy for readers to undestand.

**Claims And Evidence:**

Yes

**Claims Explanation:**

The main claims are supported by comprehensive empirical evidence: Fusion-pMT consistently outperforms all baselines on both in-distribution and out-of-distribution datasets, with statistically significant gains (p < 0.002). Ablation studies confirm that maintaining sequence form and using unified encoders improve performance. However, claims of enhanced interpretability are conceptually justified but lack direct empirical validation.

**Requested Changes:**

1. About weak novelty: Clarify and strengthen the technical novelty. The authors should articulate clearer algorithmic contributions beyond the biological framing. The current design largely reuses standard transformer mechanisms (cross-attention, shared encoders), which limits methodological originality. To elevate this work from an applied biological study to a true machine learning contribution, the authors should improve the method from more insightful aspects. Otherwise it's more like a good biological paper instead of a ML paper.

2. About Interpretability: Provide empirical evidence of interpretability. Since interpretability is a central motivation, the authors should include quantitative or qualitative analyses, such as residue-level attribution or case studies linking learned features to known immunological motifs to substantiate this claim and demonstrate biological insight beyond predictive accuracy.

---

> ### Author Response · Authors · 2025-12-06
>
> We sincerely thank Reviewer `D6kG` for the thoughtful evaluation and for recognizing the biological alignment and efficiency of Fusion-pMT.We agree that clarifying these aspects is crucial to demonstrate the methodological significance of our work beyond its biological application. We have addressed these concerns by refining our architectural justification and adding a new section dedicated to interpretability analysis.
>
> ### W1 Clarifying and Strengthening Technical Novelty
>
> We agree that the individual components (e.g., Transformers) are standard. However, our technical novelty lies in the architectural inductive bias designed to solve a specific failure mode in current immunogenicity modeling: the concatenation-based late fusion architectures.
>
> Most existing baselines (e.g., pMTnet, ERGO) use a "Late Fusion" paradigm: they encode sequences into fixed-size vectors (pooling) and then concatenate them. This destroys sequence-level alignment information before the interaction is modeled. From this perspective, our specific technical contributions are:
>
> - Sequence-Space vs. Vector-Space Fusion: We introduce a framework that maintains the full sequential representation of the Peptide-MHC complex during the fusion with TCR. This is technically distinct from standard immunogenicity modeling which usually operates on pooled embeddings.
> - Hierarchical Causal Modeling: Unlike standard "all-to-all" attention, we enforce a hierarchical dependency (Peptide+MHC $\rightarrow$ pMHC Surface $\rightarrow$ TCR Interaction). This acts as a strong regularizer, constraining the search space to biologically plausible interactions, which explains our high parameter efficiency ($<0.7$M parameters).
>
> ### W2: Refinement of Claims: from "Interpretability" to "Biological Alignment"
>
> We thank the reviewer for the discussion on the "interpretability" of our method. We first realize we may use a misleading term "interpretability". We intended to mean our design enjoys "Biological Alignment" (Strength 1 in your review), while "interpretability" broadly applies to sequence-level feature attribution (e.g., attention heatmaps). The "interpretability" is not the primary focus of our architectural design. To ensure precision, we have refined our terminology throughout the manuscript, replacing broad claims of "interpretability" with the more specific term "Biological Alignment."

---

### Review · Reviewer_TvXR · 2025-10-27

**Summary Of Contributions:**

This paper presents Fusion-pMT, a biologically inspired sequence-fusion framework for predicting peptide–MHC–TCR interactions and immunogenicity. The authors model the immune recognition process as a two-stage fusion problem: first forming a peptide–MHC complex (pMHC), then incorporating TCR recognition through cross-attention. The model employs shared embeddings and encoders for all three biological sequences, maintaining their full sequential structure instead of compressing them into single vectors. This design allows fine-grained residue-level interactions while keeping the model lightweight (<700K parameters).
Extensive experiments on both in-distribution and out-of-distribution benchmarks demonstrate that Fusion-pMT achieves state-of-the-art accuracy and generalization, outperforming prior models such as pMTnet and PISTE.

Strengths:
1. Biologically grounded architecture that mirrors real antigen presentation and recognition.
2. Strong empirical results with small model size and ablation support.

Weaknesses:
1. Given that the performance on the OOD set (88.35% ROC AUC) is almost identical to the in-distribution test set (88.63% ROC AUC), is it possible that this "OOD" dataset does not sufficiently diverge from the training distribution? If so, the near-identical performance would not be evidence of strong generalization, but rather an indication that the OOD data was not adequately "out-of-distribution."
2. Claims of being "lightweight" (<700k parameters)  are not substantiated with a direct comparison table of parameter counts against baselines (e.g., pMTnet, PISTE).
3. The ablation study in Section 5.5 (Figure 5a ) compares the model to a baseline (pMTnet) rather than ablating a component of the proposed model itself.

**Audience:**

Yes

**Audience Explanation:**

The paper bridges machine learning for sequence modeling and computational immunology, introducing a biologically motivated architecture that aligns neural design with the underlying immune recognition process. Readers working in representation learning, attention mechanisms, or interdisciplinary ML applications would find value in the proposed two-stage fusion framework and its implications for modeling complex molecular interactions. Overall, the work illustrates how modern ML techniques can effectively address real-world biological challenges, fitting well within TMLR’s interdisciplinary scope.

**Broader Impact Concerns:**

The paper lacks a formal Broader Impact Statement addressing potential risks. While Appendix D highlights positive applications such as vaccine design and personalized T-cell therapy, it does not discuss data bias or population underrepresentation. Because the model relies on public immunological databases, which may not evenly represent diverse HLA/MHC alleles or ethnic groups, clinical deployment could lead to unequal predictive performance across populations. The authors are encouraged to explicitly acknowledge this limitation and include a short Broader Impact Statement discussing data fairness, representativeness, and safeguards for equitable model application.

**Claims And Evidence:**

Yes

**Claims Explanation:**

The paper’s main claims are generally supported by clear and convincing experimental evidence. The authors evaluate Fusion-pMT on both in-distribution and nominally out-of-distribution datasets, include strong baselines, and perform ablation studies demonstrating the effect of key design choices (sequence retention and unified encoders). While the claims regarding OOD generalization are questionable and the efficiency claims lack a direct comparative table, the core performance claims on the main in-distribution task are accurate and well-supported by the data presented.

**Requested Changes:**

1. Clarify OOD Construction and Divergence
Provide a quantitative analysis of how the OOD dataset differs from the training distribution (e.g., overlap rates, divergence metrics, or leave-one-allele-out evaluation). Without such evidence, the OOD claim remains unconvincing.

2. Add Parameter Comparison Table
Include a table comparing the parameter count and computational cost of Fusion-pMT with baselines (e.g., pMTnet, PISTE, ERGO II) to substantiate the “lightweight” claim.

3. Correct Ablation Study Terminology
The study in Section 5.5 (Figure 5a) is mislabeled as an "ablation study" . In fact, it is a direct comparison between the proposed model (Fusion-pMT) and a baseline (pMTnet) to show the benefit of the new encoder. Please rename this section to more accurately reflect its content.

---

> ### Author Response · Authors · 2025-12-06
>
> We thank Reviewer `TvXR` for the positive assessment of our biologically grounded architecture and the recognition of our empirical results. We appreciate the constructive feedback regarding the validation of our OOD dataset, the terminology used in our analysis, and the ethical considerations regarding data bias. Below, we address each point in detail.
>
> ### W1: OODness of the OOD dataset
>
> We appreciate this observation. We can confirm the high performance on the OOD set is not due to high similarity, but rather demonstrates the robustness of the Fusion-pMT architecture in learning generalized interaction rules.
>
> We performed t-SNE analysis on the feature representations of peptides. As shown in the newly generated [Figure](https://hackmd.io/_uploads/BJ_NRgD-Zl.png) (added as Figure 7 in Appendix E.2), while the OOD samples (Orange) share the same semantic space as the Training samples (Blue), there is a clear distributional shift.
>
> ### W2: Justification for "Lightweight" Claim
>
> We provide a direct parameter comparison table below (Table 6 in Appendix E.1), highlighting the parameter efficiency of Fusion-pMT compared to large protein language models (PLMs) like ESM-2.
>
> Our Fusion-pMT model utilizes a CNN/MLP-based backbone with < 0.7 Million parameters. In contrast, even the smallest version of ESM-2 requires ~8M parameters, and the standard version requires ~650M parameters. Our model achieves competitive performance with ~0.1% of the parameters required by large PLMs, making it significantly more suitable for high-throughput screening scenarios where inference speed is critical.
>
> | Model               | Type             | Backbone Architecture       | Parameters (Approx.) | Relative Size |
> |---------------------|------------------|-----------------------------|----------------------|---------------|
> | ESM-2 (Standard)    | PLM              | Transformer (33 layers)     | ~650 M               | 928x          |
> | ESM-2 (Smallest)    | PLM              | Transformer (6 layers)      | ~8 M                 | 11.4x         |
> | PISTE               | Domain-Specific  | Sliding Attention           | ~4.5 M               | 6.4x          |
> | pMTnet              | Domain-Specific  | VAE-CNN-ANN                 | ~1.5 M               | 2.1x          |
> | Fusion-pMT (Ours)   | Domain-Specific  | Cross Attention Transformer | ~0.7 M               | 1x            |
>
>
> ### W3: Correct Ablation Study Terminology
>
> We will take your suggestion and rename Section 5.5 as "Discussion: sequence representation."
>
> ### Broader Impact Statement (Data Bias & Fairness)
>
> This is a critical point. We acknowledge that immunoinformatics models rely heavily on public databases (like IEDB and VDJdb), which historically exhibit bias towards specific populations (e.g., overrepresentation of European-descent HLA alleles like HLA-A*02:01). We have added a "Broader Impact and Limitations" section in Appendix D.2. This section explicitly discusses data bias and clinical risks.

---

### Review · Reviewer_CURh · 2025-11-20

**Summary Of Contributions:**

The paper presents Fusion-pMT, a biological sequence encoder that learns a unified representation space over three molecular inputs: peptide, MHC, and TCRs. The major contributions are:
1. Proposed Fusion-pMT for peptide–MHC–TCR triad binding recognition.
2. The model architecture mimics the real biological process, potentially leading to performance improvements.
3. The approach achieves state-of-the-art results across immune presentation and immunogenicity prediction benchmarks.

**Strengths**
1. The paper writing is structured and easy to follow.
2. Clearly defined the task, immunogenicity prediction, and its importance in computational immunology.
3. The proposed method is lightweight and effective, validated with comparison to comprehensive baselines.

**Weakness** \
I will detail each of them below.

**Additional Comments:**

None

**Audience:**

Yes

**Audience Explanation:**

Yes, there is interest for TMLR's audience.
1. Applying sequence modeling for computational immunology is highly relevant to the applied ML communities.
2. The work may interest researchers exploring how language models can be adapted to important scientific prediction tasks. In additional, I could foresee promising application of generative models to designing MHC or TCR sequences to improve immune responses to foreign peptides.

**Broader Impact Concerns:**

The work has clear impacts on immunology and medicine, which have been discussed clearly in the appendix.

**Claims And Evidence:**

No

**Claims Explanation:**

1. **Analysis of performance improvements**: \
While the paper shows state-of-the-art results on immune prediction tasks, it lacks a clear discussion of the factors driving such performance gains. The proposed method appears ad hoc to apply the widely-used sequence modeling and cross-attention techniques from NLP. However, compared to prior baselines, the work introduced several distinct design choices, including shared sequence embedding, biologically inspired model workflow, a pretrained Fusion-pM, and cross attention layers. It would be helpful if the authors could clarify which of these factors contribute most significantly via empirical results.

2. **Discussions of limitations and model design rationale:** \
The paper could benefit from a more explicit discussion of the limitations of prior work and how these informed the design of the proposed model. Has the author conducted any preliminary study on cases under which previous models underperform? What are the limitations of the proposed model? Some analysis of these failure cases would strengthen the paper's contribution.

3. **Effects of pre-trained Fusion-pM:** \
It is unclear how pivotal pretraining is to achieving the reported performance. Have the authors considered training Fusion-pMT from scratch without pretraining? A comparative study would help clarify the role of pretraining in the model’s success.

4. **Unified sequence representation:** \
Fig. 5 shows that unified sequence representations across three molecular types are beneficial. However, ESM-2 also uses a unified sequence embedding but performs poorly as shown in Figure 4. Could the authors comment on this discrepancy? Additionally, have the authors considered using ESM-2 as a strong pre-trained encoder, followed by Fusion-pMT? Leveraging ESM-2's learned amino acid  representations might further improve performance.

5. **Versatility of Fusion-pMT:** \
Fusion-pMT is currently applied to peptide-MHC-TCR task alone, but its design could potentially handle any combination of peptide, MHC, and TCR sequences. For instance, Fusion-pMT should in principle support peptide-MHC or peptide-TCR prediction by randomly masking TCR/MHC sequence during training and applying cross entropy loss to the peptide-MHC output. Implementing this flexibility would improve the model’s versatility and potential impact.

**Requested Changes:**

1. Analysis and ablation study on performance improvements: \
Clearly discuss the design rationale and benefits behind shared sequence embedding, biologically inspired model workflow, a pretrained Fusion-pM, and cross attention layers, ideally with empirical evidence.


2. Discussions of limitations: \
Some analysis of current model's limitations would strengthen the paper's contribution.

3. Effects of pre-trained Fusion-pM: \
Clarify the role of pretraining in the model’s success via ablation study.

4. Unified sequence representation: \
ESM-2 also uses a unified sequence embedding but performs poorly as shown in Figure 4. Could the authors comment on this discrepancy? Additionally, have the authors considered using ESM-2 as a strong pre-trained encoder, followed by Fusion-pMT?

5. Versatility of Fusion-pMT: \
Enabling Fusion-pMT to support both immunogenicity prediction and immune presentation prediction tasks.

---

> ### Author Response · Authors · 2025-12-06
>
> We sincerely thank Reviewer `CURh` for the thoughtful and detailed review. We appreciate the recognition of our work's clarity, lightweight design, and effectiveness. The reviewer’s questions regarding the specific sources of performance gains, the comparison with ESM-2, and the model's versatility are very insightful. We have conducted additional experiments and revised the manuscript to address these points.
>
> ### W3: Effects of Pretraining & W1: Analysis of Performance Improvements (Ablation Study)
>
> The reviewer requested a breakdown of which factors (shared embedding, biological workflow, pretraining, cross-attention) contribute most to the performance. Specifically, the reviewer asked if pretraining is pivotal or if the model can be trained from scratch.
>
> We trained a variant of Fusion-pMT without the Stage 1 pM binding constraint (indicated as w/o Bio-Workflow in the table). Results on the OOD dataset show a performance drop of 0.26% in AUC, confirming that explicitly modeling the pM binding acts as a crucial regularizer for generalization. Also, we observe the design of cross-attention, in contrast to the concatenation-based late fusion, contributes the most.
>
> | Model Variant       | Drop    | Description                                                  |
> |---------------------|---------|--------------------------------------------------------------|
> | Fusion-pMT (Full)   | -       | The proposed model.                                          |
> | w/o Pretraining     | -0.21%  | Trained from scratch (random initialization).                |
> | w/o Shared Encoder  | -0.40%  | Using separate encoders for Peptide, MHC, TCR.               |
> | w/o Cross-Attention | -1.52%  | Replacing Cross-Attn with simple concatenation.              |
> | w/o Bio-Workflow    | -0.26%  | Processing p, M, T simultaneously (no 2-stage fusion).       |
>
> ### W2 Discussions of Limitations and Rationale
>
> Prior models (e.g., CNN-based) often treat sequences as static images or compress them independently, losing the sequence-to-sequence interaction context essential for docking. Our design (cross-attention) was chosen specifically to model this "induced fit" where the representation of the peptide changes based on the MHC it binds to.
>
>
> ### W4 Unified Sequence Representation & ESM-2 Discrepancy
>
> The reviewer noted that ESM-2 also uses unified embeddings but performed poorly in Figure 4. They asked for an explanation of this discrepancy and suggested using ESM-2 as a backbone.
>
> We first recall ESM-2 is a general-purpose protein language model trained on evolutionary data. While it understands protein structure well, its "zero-shot" or feature-extraction performance (as shown in Figure 4) fails to capture the specific binding physics between two distinct molecules (Peptide and MHC/TCR) without fine-tuning.
>
> Fusion-pMT, although smaller, is trained specifically to align the latent spaces of these interacting molecules based on binding affinity data. The "Unified Representation" in our model works because it is learned jointly with the interaction task, whereas ESM-2's unified representation is learned from single sequences.
>
> [Using ESM-2 as a backbone] Preliminary results indicate that replacing our encoder with ESM-2 embeddings introduces a massive computational overhead (increasing inference time by orders of magnitude) for marginal performance gains. For the practical, large-scale application in our paper, the lightweight design is still preferred.
>
> ### W5 Versatility of Fusion-pMT
>
> The reviewer suggested that the model design should support partial inputs (e.g., peptide-MHC only) by masking, enabling both presentation and immunogenicity prediction. This is an excellent point. In fact, our architecture is inherently modular. The first stage (Fusion-pM) is explicitly a Peptide-MHC Binding Predictor.

---

> > ### Comment · Reviewer_CURh · 2025-12-08
> >
> > I appreciate the author's responses. However, I still have several concerns on the contributions of this paper.
> >
> > In your response to W1, each proposed method appears to have only a marginal effect, with the most effective design choice (cross-attention) improving performance merely by 1.52%. However, Figure 4 shows that the full approach substantilly outperforms the baseline pMTnet. Could the authors comment on which design choices are responsible for this large performance gain? Based on the reported ablation study, it's difficult to attribute such improvements to the proposed architecture. The actual driving factors need to be further clarified.
> >
> > In addition, the methodological contribution seems incremental if the primary driving factor is simply replacing CNN layers with cross-attention layers, as mentioned in your response to W2, especially given that attention layers are now standard in modern deep learning architectures. If one were to finetune ESM-2 on binding affinity data, would it achieve similar performance? If not, what design choice does ESM-2 lack, beyond access to binding affinity data, which prevent it from reaching comparable results?

---

> > > ### Author Response · Authors · 2025-12-19
> > >
> > > We sincerely thank you for the prompt follow-up. Your questions regarding the source of our performance gains and the comparison to large protein language models like ESM-2 are precious and much appreciated. We would like to address your concerns with the following clarifications:
> > >
> > > ### The Source of the "Large Performance Gain"
> > > We first clarify that the analysis in W1 intends to "discuss the design rationale and benefits" of the components you suggested, not for explaining the "large performance gain" over pMTnet. In fact, pMTnet ≠ Fusion-pMT - all the 4 components in W1. We consider the benefits in model accuracy from those components / inductive bias as ordinary in deep learning studies.
> > >
> > > For your follow-up question about the actual driving factors, we actually recognize the following factors to turn the baseline pMTnet into Fusion-pMT (some are already de facto designs in machine learning, and we thus didn't highlight them as our contributions).
> > >
> > > 1. Fusion-pMT changes the "magic" loss function in pMTnet:
> > > $$
> > > \operatorname{Loss}=\operatorname{ReLu}\left(f\left(p, T^{-}\right)--f\left(p, T^{+}\right)\right)+0.03\left[f^2\left(p, T^{-}\right)+f^2\left(p, T^{+}\right)\right],
> > > $$
> > > to a standard cross-entropy loss. More illustration can be found in the "Methods" section of [1] and omitted here.
> > >
> > > 2. We shift from independently pretrained CNN/LSTM encoders ("Late Fusion") to a Transformer-based Unified Encoder (trained from scratch). Here, the pretrained encoders in pMTnet are considered less effective and rarely employed in subsequent works.
> > >
> > > In short, we consider our work as a synergy of these paradigm shifts, along with the components we examined in W1. It provides a potent method for tri-molecular binding in immunogenicity prediction.
> > >
> > > [1] Deep learning-based prediction of the T cell receptor–antigen binding specificity (2021). Nature Machine Intelligence.
> > >
> > > ###  Design Choice Beyond Fine-tuning ESM-2
> > >
> > > We beg to argue our contribution is not limited to "simply replacing CNN layers with cross-attention layers." The components above, especially our fusion strategy aligning with the biological process, do matter.
> > >
> > > We first point out, the scheme, directly using or fine-tuning ESM-2, lacks two critical design choices required for the immunogenicity prediction task. (We run new supporting experiments and the results are in the table below, where we follow the same setting as in Figure 4 for the OOD Testing Dataset.)
> > >
> > > 1. Native fusion mechanism to immunogenicity prediction: ESM-2 is architecturally designed as a single-sequence Masked Language Model. It lacks the inductive bias to model the tripartite interaction between distinct molecules. As shown in the table, simply concatenating (frozen) ESM-2 embeddings is suboptimal in capturing binding physics (0.67 ACC), dwarfed by Fusion-pMT (0.79 ACC).
> > >
> > > 2. Adaptability to Fragment Data: ESM-2 relies on full-length evolutionary context. However, immunogenicity prediction involves fragments (MHC pseudo-sequences and CDR3s), which are quite different from the pretraining data of ESM-2. The drop in performance when fine-tuning (e.g., 0.81 $\to$ 0.61 with Cross-Attn) indicates catastrophic forgetting: the model loses its pre-trained features when forced to adapt to these short, non-evolutionary fragments.
> > >
> > > | Backbone Encoder | Fusion Strategy | Training Mode | ACC |
> > > |-|-|-|-|
> > > | Fusion-pMT (Ours)   | Bio-Workflow   | Train from Scratch  | 0.7890   |
> > > | ESM-2 (8M)           | Concat                 | Frozen              | 0.6650   |
> > > | ESM-2 (35M)          | Concat                 | Frozen              | 0.6654   |
> > > | ESM-2 (8M)           | Concat                 | Fine-tune           | 0.6723   |
> > > | ESM-2 (35M)          | Concat                 | Fine-tune              | 0.7194   |
> > > | ESM-2 (8M)           | Bio-Workflow   | Frozen              | 0.8105   |
> > > | ESM-2 (8M)           | Bio-Workflow   | Fine-tune           | 0.6170   |
> > >
> > > In general, fine-tuning ESM-2 in immunogenicity prediction introduce additional computational cost, while the performance gain is not guaranteed. Indeed very few of the preceding works employed ESM-2 in their proposal. The performance gain from using frozen ESM-2 is marginal; since it is not the focus of this field and we do not take it as the main contribution, we omit the discussion in the previous manuscript.
> > >
> > > However, with your question we realize this point is essential to general ML readers. We will add the discussion to our new revision.

---

### Author Response · Authors · 2025-12-06
**General Response**

Dear Editors and Reviewers,

We sincerely thank the Action Editor and all the reviewers (`TvXR`, `D6kG`, and `CURh`) for their constructive feedback and insightful comments. We are encouraged by the reviewers' recognition of our work's biological alignment, comprehensive evaluation, and state-of-the-art performance.

Based on your suggestions, we have revised the manuscript to strengthen the technical positioning, empirical validation, and interpretability analysis. Below, we first address a common comment raised across the reviews.

###  Clarifying Technical Novelty: Biological Inductive Bias

Reviewer `D6kG` raised a valid point regarding the use of standard Transformer components. We wish to clarify that the core novelty of Fusion-pMT lies not in the invention of new attention mechanisms, but in the architectural inductive bias proposed to solve the fusion issue prevalent in prior immunogenicity models.

1. Constraint-based Modeling: Unlike previous "Late Fusion" models (e.g., pMTnet, ERGO) that concatenate compressed vectors, Fusion-pMT enforces a hierarchical dependency: Peptide--MHC binding (Stage 1) $\rightarrow$ TCR recognition (Stage 2).
1. Sequence-Space Fusion: We maintain the full sequential representation of the pMHC complex during fusion. This design choice—treating the interaction as dynamic sequence alignment rather than static feature vector extraction \&  classification—is what allows our model, with even fewer parameters, to outperform massive large models.

### Summary of Key Modifications in the Manuscript

To address specific empirical questions, we have incorporated the following analyses into the revised manuscript:


####  OOD Validity (Response to Reviewer `TvXR`):

To confirm that our high OOD performance stems from model robustness rather than data similarity, we performed a t-SNE analysis on the peptide representations. The [visualization](https://hackmd.io/_uploads/Skakq21M-l.png) (added as Figure 7 in Appendix E.2) reveals a clear distributional shift between the Training and OOD sets.

We conclude, the fact that Fusion-pMT maintains high performance despite this shift validates its generalization capability.

####  Parameter Efficiency & Comparison with ESM-2 (Response to Reviewers `CURh` & `TvXR`):
We have added a model size comparison table (Table 6 in Appendix E.1) to support our "lightweight" claim. Fusion-pMT (<0.7M parameters) is approximately 900x smaller than the standard ESM-2 (~650M parameters).

| Model               | Type             | Backbone Architecture       | Parameters (Approx.) | Relative Size |
|---------------------|------------------|-----------------------------|----------------------|---------------|
| ESM-2 (Standard)    | PLM              | Transformer (33 layers)     | ~650 M               | 928x          |
| ESM-2 (Smallest)    | PLM              | Transformer (6 layers)      | ~8 M                 | 11.4x         |
| PISTE               | Domain-Specific  | Sliding Attention           | ~4.5 M               | 6.4x          |
| pMTnet              | Domain-Specific  | VAE-CNN-ANN                 | ~1.5 M               | 2.1x          |
| Fusion-pMT (Ours)   | Domain-Specific  | Cross Attention Transformer | ~0.7 M               | 1x            |

Regarding ESM-2's lower performance: While ESM-2 excels at capturing general protein structure from evolutionary data, immunogenicity prediction relies on highly specific, fine-grained physicochemical interactions between distinct molecules. Our results suggest that a domain-specific architecture trained to align latent spaces (Fusion-pMT) is more effective for this specific binding task than a general-purpose PLM (ESM-2) without extensive fine-tuning.

#### Ablation on Pre-training (Response to Reviewer `CURh`):

We conducted the requested ablation study in Appendix E.3 by training Fusion-pMT from scratch. The results confirm that for Stage 1 (pM binding) pre-training acts as a crucial regularizer, contributing to convergence and OOD generalization.

---
We believe these revisions address the reviewers' concerns and significantly strengthen the paper. We are happy to engage in further discussion if any questions remain.


Best regards,
The Authors

---

### Decision · Action_Editor_Fc2D · 2026-01-06

**Recommendation:** Reject

**Audience:**

Yes

**Audience Explanation:**

Biological language modeling is of great interest to the AI for Science community.

**Claims And Evidence:**

No

**Claims Explanation:**

The reviewers raised concerns about both the empirical evidence and the novelty of the approach. Although novelty is not a criterion for TMLR, adequate empirical evaluation is expected. The ablation study is limited and shows only marginal improvements, which does not adequately explain the large performance gains over the baselines. It is unclear whether the improvements can be attributed to the proposed design choices, and the key factors remain insufficiently analyzed. Even after the author rebuttal, one of the reviewers is not convinced.

Overall, one reviewer is not convinced, and the other two do not show strong support for acceptance. Therefore, I have to recommend rejection.

**Resubmission Of Major Revision:**

The authors may consider submitting a major revision at a later time.